# A Smart Hyperthermia Nanofiber-Platform-Enabled Sustained Release of Doxorubicin and 17AAG for Synergistic Cancer Therapy

**DOI:** 10.3390/ijms22052542

**Published:** 2021-03-03

**Authors:** Lili Chen, Nanami Fujisawa, Masato Takanohashi, Mazaya Najmina, Koichiro Uto, Mitsuhiro Ebara

**Affiliations:** 1Research Center for Functional Materials, National Institute for Materials Science (NIMS), 1-1 Namiki, Tsukuba, Ibaraki 305-0044, Japan; CHEN.Lili@nims.go.jp (L.C.); nfujisawa.tokyo@gmail.com (N.F.); takanohashi1213.lab@gmail.com (M.T.); NAJMINA.Mazaya@nims.go.jp (M.N.); UTO.Koichiro@nims.go.jp (K.U.); 2Graduate School of Pure and Applied Sciences, University of Tsukuba, Tsukuba, Ibaraki 305-0006, Japan; 3Department of Materials Science and Technology, Tokyo University of Science, Tokyo 125-8585, Japan

**Keywords:** cancer thersapy, hyperthermia, MNPs, DOX, HSPs inhibitor, nanofiber

## Abstract

This study demonstrates the rational fabrication of a magnetic composite nanofiber mesh that can achieve mutual synergy of hyperthermia, chemotherapy, and thermo-molecularly targeted therapy for highly potent therapeutic effects. The nanofiber is composed of biodegradable poly(ε-caprolactone) with doxorubicin, magnetic nanoparticles, and 17-allylamino-17-demethoxygeldanamycin. The nanofiber exhibits distinct hyperthermia, owing to the presence of magnetic nanoparticles upon exposure of the mesh to an alternating magnetic field, which causes heat-induced cell killing as well as enhanced chemotherapeutic efficiency of doxorubicin. The effectiveness of hyperthermia is further enhanced through the inhibition of heat shock protein activity after hyperthermia by releasing the inhibitor 17-allylamino-17-demethoxygeldanamycin. These findings represent a smart nanofiber system for potent cancer therapy and may provide a new approach for the development of localized medication delivery.

## 1. Introduction

Breast cancer is the most common malignancy diagnosed in women worldwide and the second leading cause of cancer-related deaths in women [1,2]. Despite recent advances in early diagnosis and treatment, clinical applications are still limited to single therapeutic approaches. Therefore, to achieve preferable anticancer efficacy, multimodal therapies containing two or more therapeutic modalities have been extensively explored in clinical settings in the past decades. The combination therapy approach has been the preferable mode of treatment for cancer patients to significantly reduce tumor size; for example, chemotherapy is combined with gene therapy, radiotherapy, and thermal therapy [3] (Scheme 1). Thermal therapy, also known as hyperthermia, is a mode of cancer treatment that can damage and kill cancer cells, which is employed based on evidence that indicate that cancer cells are more sensitive than normal cells to high temperatures (43–45 °C) [4,5,6]. Hyperthermia can also enhance the effects of certain anticancer drugs such as salinomycin [7,8], curcumin [9], paclitaxel (PTX) [10,11] and doxorubicin (DOX) [12,13,14], increasing the susceptibility of some cancer cells to them.

Although combination therapy is an effective strategy to improve anticancer efficiency through hypothermia and chemotherapy, cancer cells can activate cytoprotective and anti-apoptotic pathways, such as heat shock response that may impart thermoresistance to the cells upon applying hyperthermia [15]. Generally, the heat shock response can considerably induce the overexpression of heat-shock proteins (HSPs), which reaches the highest level within a few minutes to 30 min [16]. When the cells are thermally stimulated, increased production of Hsp90 as a member of chaperone molecules that regulate the intracellular expression, function, and stability of client proteins can particularly stabilize many oncogenic proteins such as mutantp53, Raf-1, HER/ErbB2, Akt, Bcr-Abl, and steroid receptors, thereby leading to thermoresistance and anti-apoptosis [17,18,19]. However, Hsp90 inhibitors can bind to Hsp90 and cause proteasomal degradation of Hsp90 client proteins, inhibit its protein activity, and enhance the effect of hypothermia. Meanwhile, inhibitors of Hsp90 as a type of molecular targeted drug can also exert the effect of molecularly targeted therapy [20].

Hsp90 inhibitors are molecular targeting drugs that target the chaperone protein of Hsp90 in the cytoplasm, cause the conformational change of Hsp90 and degrade the protein substrate by binding to the regulatory site of Hsp90, thus preventing cell growth, proliferation, and signal transmission [21]. 17-N-allylamino-17-demethoxygeldanamycin (17AAG) as the first Hsp90 inhibitor for clinical trials for cancer therapy can reduce thermoresistance in hyperthermia induced by various stimuli such as magnetic fields, and thus induce distinct apoptosis for achieving molecularly targeted therapy [20,22,23,24]. Notably, the binding affinity of 17AAG for Hsp90 of multi-chaperone complexes with high ATPase activity in cancer cells is 100-fold higher than that of the inactive Hsp90 from normal cells [25]. Therefore, these properties give 17AAG an advantage over other single signal-blocking antitumor drugs.

The repair and reconstruction of tissue defects after tumor resection are of major importance for long-term successful healing in cancer therapies [26]. Therefore, recent advances in early diagnosis and treatment of cancer have called for more effective cancer treatment approaches [27,28]. This study presents new insights into an effective locally implantable system for enhancing the efficacy of combination cancer treatment.

## 2. Results and Discussion

### 2.1. Fabrication of Nanofiber Meshes

PCL is commonly used as a biodegradable material for biomedical applications and approved by the US Food and Drug Administration. Owing to its biocompatibility, high hydrolysis activity, and desirable processability, PCL has been extensively utilized in drug delivery, tissue engineering scaffolds, and implant materials [29,30]. Electrospinning is a straightforward and versatile technique used to fabricate fibers with nanoscale diameters from a variety of polymeric materials [31,32,33]. We have previously reported PCL-based nanofiber meshes incorporating various types of therapeutic agents such as inactivated viruses [34], immune-modulating agents [35], and chemotherapeutic drugs [10,36] for cancer therapy. In this study, PCL nanofibers containing MNPs, DOX, and 17AAG were fabricated through blend electrospinning. Figure 1A shows the optical microscopy and SEM images of a variety of electrospun PCL-based nanofiber meshes. These nanofibers showed uniform diameter distributions without any beaded morphology. Figure 1B shows that the average diameter of various PCL-based nanofibers ranged from 400 nm to 700 nm, where the diameters of PCL nanofiber and MNP/DOX/17AAG-PCL nanofiber were 610 ± 87 nm and 690 ± 73 nm, respectively.

### 2.2. Heating of MNPs within Nanofiber Meshes

Magnetic hyperthermia is a type of thermal therapy that can be employed to treat tumors using heat generated by magnetic nanoparticles with exposure to an AMF [37]. It can precisely deal with deep tumors in organs owing to the excellent tissue penetration ability of AMF [38,39,40,41]. Therefore, it is vital to detect the heating potential of MNPs within nanofiber meshes exposed to AMF.

As shown in Figure 2A, the infrared thermal images of PCL, MNP-PCL, and MNP/DOX/17AAG-PCL nanofiber meshes loaded with 12.0 mg of MNPs under AMF irradiation for 15 min. Compared to PCL nanofiber mesh with no significant change in temperature, the temperature of an AMF-exposed mesh of MNP-PCL and MNP/DOX/17AAG-PCL MNPs increased from 25.8 °C and 25.9 °C to 44.1 °C and 43.8 °C, respectively. Figure 2B shows the change in temperature of the MNP-PCL nanofiber mesh with different MNP contents in the AMF application for 15 min. The nanofibers caused the temperature to increase to 32.0, 43.6, and 54.8 °C in the meshes with 2, 12, and 24 mg of MNPs, respectively. Only small changes in temperature were measured for the mesh with 2 mg of MNPs, while the fiber meshes with 12 and 24 mg of MNPs showed significant changes within 5 min.

Additionally, the time-dependent temperature changes of the PCL, MNP-PCL, and MNP/DOX/17AAG-PCL nanofiber meshes and the MNP-PCL nanofiber meshes with different amounts of MNPs during AMF irradiation were analyzed. From Figure 2C,D, the temperature of the nanofiber meshes showed a sharp increase during the first 90 s of AMF application before reaching a plateau. From Figure 2C, there were no significant changes in the temperature of the PCL mesh within 300 s. In contrast, the temperature of meshes loaded with MNPs increased to above 43 °C at 300 s. Figure 2D shows that the temperature increased to 32.3, 43.7, and 55 °C for the meshes with 2, 12, and 24 mg of MNPs, respectively. Thus, the heating rates of MNPs within the nanofiber meshes varied with concentration and time. Additionally, the heat generation behavior of the MNP-incorporated nanofiber mesh was hardly affected in the presence of DOX and 17AAG.

In general, magnetic fields have some effects on tissues and organ systems, such as induced eddy currents in tissues, which may cause carbonization or necrosis of healthy tissues. The strengths of magnetic fields are limited in clinical applications [42]. Brezovich et al. experimentally determined the safety frequency threshold of AMF to be 100–300 kHz [43]. In this study, we used AMF of 166 kHz, the intensity of which is relevant for clinical applications.

### 2.3. DOX and 17AAG Release Behavior In Vitro

For nanoparticle platforms, it is difficult to achieve long-term release because the loading capacity of a drug in a particle is limited. Therefore, repeated administration is required for nanoparticle systems [44]. Compared with these nanoparticle drug-delivery platforms, nanofiber meshes possess higher drug-loading capacity and longer release potential, which is attributed to the unique features of nanofiber materials. This study focused on demonstrating the sustained release of DOX and 17AAG from nanofiber meshes. Additionally, the release kinetics of the drug from the nanofibers were assessed at 37 °C in PBS buffers at different pH levels to imitate normal extracellular fluid or blood (pH 7.4) and the tumor environment (pH 6.5) [45].

As shown in Figure 3, the MNP/DOX/17AAG-PCL nanofiber showed sustained release of DOX and 17AAG over 30 days. In addition, more than 30% of loaded DOX and 17AAG contents were released from MNP/DOX/17AAG-PCL with and without AMF application at 30 days. This indicates that AMF irradiation itself did not affect the DOX and 17AAG release. Moreover, the release of DOX and 17AAG from the MNP/DOX/17AAG-PCL nanofiber showed slight pH-dependence, indicating an increase in the release rate at pH 6.5, rather than 7.4, in the release medium. This is because lower pH leads to increased solubility of DOX, which induces easier drug release [46]. Thus, the pH-sensitive release behavior of DOX and 17AAG from the MNP/DOX/17AAG-PCL nanofiber was favorable for DOX and 17AAG in improving the therapeutic effect for cancer.

### 2.4. Effect of Heat on the Efficiency of DOX and 17AAG

Clinical studies have demonstrated that magnetic nanoparticle (MNP)-based hyperthermia enhances the cytotoxicity of some chemotherapeutic drugs [47,48], such as salinomycin [7], curcumin [9], DOX [12], and 17AAG [49]. Mild hyperthermia treatment (43–45 °C) can improve the sensitivity to drugs in the entire tumor by increasing cell spacing and permeability [50,51,52,53]. To investigate the maximized synergistic effect of combination therapy of hyperthermia with chemotherapeutic drugs (DOX and 17AAG) toward heat-sensitive MCF-7 cells, cell viability was evaluated at different DOX and 17AAG concentrations at different heating times at 43 °C.

As demonstrated in Figure 4A, the cell viability in the absence of DOX showed a significant decrease with increased heating time. It is demonstrated that a single hyperthermic treatment can induce cell death in MCF-7 cells. This may be attributed to heating, which can induce a thermal denaturation effect on protein and DNA in cells. In particular, hypoxic tumor cells are more sensitive to heating than normal tissue cells. In addition, the effect of DOX on MCF-7 cells was also showed notably and survival cell numbers were decreased with the increase of DOX concentration. Interestingly, the anticancer effect of DOX was synergistically enhanced by a combination of hyperthermia at 43 °C for 15 min. Compared to the cells without heating, the cell viabilities decreased from 88.7% and 54.5% to 30.3% and 3.2% with 0.1 and 20 μg/mL of DOX, respectively. A similar decrease in the survival of MCF-7 cells was observed, as shown in Figure 4B. This may be attributed to the fact that hyperthermia increases the permeability of the cancer cell membrane, thus aiding drug penetration [12]. Moreover, heating improves cross-linking of the drug with DNA and inhibits the repair of cancer cells to drug damage, thus enhancing the anti-cancer effect of the drug [54,55].

Furthermore, the effect of heating time on the synergistic efficiency of the combination of hyperthermia with DOX (0.1 μg/mL) and 17AAG (0.1 μg/mL) was further investigated. As illustrated in Figure 4C,D, cell viability decreased distinctly with increased heating time. Cytotoxicity significantly dropped with heating for more than 15 min in comparison with the control. After heating for 90 min, the viability of MCF-7 cells treated with DOX and 17AAG was reduced to 22.5% and 23.7%, respectively. Nevertheless, the cells heated for 120 min displayed similar cytotoxicity to heating for 90 min. These results indicate that the anticancer effect of DOX and 17AAG was enhanced by the combination with hyperthermia at 43 °C and displayed a time-dependent relationship.

### 2.5. Synergistic Anticancer Effects

Hyperthermia, a common clinical treatment method, can enhance the effect of DOX by increasing the susceptibility of cancer cells; however, hyperthermia-induced thermoresistance of cancer cells may be caused by heat shock response. 17AAG is a potent Hsp90 inhibitor that can effectively target Hsp90, thus reducing thermoresistance caused by hyperthermia [20,22,23,24]. More interestingly, 17AAG also has tumoristatic effects as a targeted drug and has been demonstrated to have a synergistic effect with chemotherapy [56,57], benefiting from inhibition of most of the pro-survival and angiogenic signaling molecules elevated by hyperthermia. Here, “All in one” nanofiber mesh based on a synergistic anticancer effect from hyperthermia with DOX and 17AAG was analyzed by the viability of MCF-7 cells.

As demonstrated in Figure 5, the nanofiber mesh (NFM) and only-AMF treatment group maintained cell viability higher than 95%, indicating that the blank nanofiber mesh and AMF were non-toxic to MCF-7 cells. The DOX and 17AAG loaded nanofibers with MNPs displayed higher cytotoxicity in comparison with DOX-NFM and 17AAG-NFM. On the other hand, the MNPs/DOX/17AAG-NFM also showed higher cytotoxicity than NFM loaded with MNPs alone (MNPs-NFM) and the DOX/17AAG-NFM absent of MNPs. This result was in agreement with the heat effect on the efficiency of DOX and 17AAG (Figure 4), and further emphasized that hyperthermia can improve the cytotoxicity of DOX and 17AAG. The cell viability of MNPs/DOX/17AAG-NFM was 11.6%. It was found that MNPs/DOX/17AAG-NFM killed cells significantly compared to the control because of the combined effect of hyperthermia with DOX and 17AAG. Thus, these results demonstrated that the MNPs/DOX/17AAG nanofiber mesh could dramatically increase cytotoxicity in MCF-7 cells through the combined anticancer effect. Therefore, this platform may be further developed as an implantable delivery system in breast cancer therapy.

## 3. Materials and Methods

### 3.1. Materials

Poly(ε-caprolactone) (PCL, Mw = 80 kDa) (>99.0%), 1,1,1,3,3,3-hexafluoro-2-propanol (HFIP) (>99.0%), and DOX (>95.0%) were purchased from Tokyo Chemical Industry Co., Ltd. (TYO, Japan). Iron (III) oxide nanopowder (less than 50 nm particle size) and 17AAG (≥98%) were obtained from Sigma-Aldrich Japan (TYO, Japan). Modified Eagle’s Medium (MEM), trypsin, penicillin, and streptomycin were obtained from Nacalai Tesque, Inc. (Kyoto, Japan). Fetal bovine serum (FBS) was purchased from Tocris Bioscience Inc. (Minneapolis, MN, USA). Alamar blue reagent was obtained from TREK Diagnostics (Cleveland, OH, USA). MCF-7 human breast cancer cells were purchased from the American Type Culture Collection (Manassas, VA, USA).

### 3.2. Fabrication and Characterization of Nanofiber Mesh

The PCL nanofiber mesh was fabricated according to our previously published procedure [10]. Briefly, PCL was dissolved in HFIP at a concentration of 20% (*w*/*v*) to prepare an electrospinning solution. The MNPs, DOX, and 17AAG were dissolved in the PCL solution at concentrations of 30% (*w*/*v*), 0.75%, and 0.75% (*w*/*w*), respectively. For the drug-loaded nanofibers, DOX or 17AAG was dissolved in HFIP and then added to the PCL solution prior to electrospinning and stirred thoroughly to form a homogeneous solution. The PCL-based nanofibers were produced through an electrospinning system (Nanon-01A, MECC Co., Ltd., Japan). A positive voltage of 20 kV is applied to the polymer solution to overcome the liquid surface tension and enable the formation of a polymer jet. The nanofibers were collected on a collector plate 8 cm away from the syringe needle. The flow rate was set at 1.0 mL/h and the experiment was performed at room temperature.

The morphology of the nanofibers was observed by scanning electron microscopy (SEM, SU8000, Hitachi High-Technologies Corporation, Tokyo, Japan) using secondary electrons after Pt coating the surface of the nanofibers. The diameter of the nanofibers was determined using Image J and plugin diameter J.

### 3.3. Heating Profiles for Nanofiber Mesh

The heat generation properties of the MNPs in the nanofiber mesh were explored by alternating magnetic field (AMF) irradiation. Nanofiber meshes were placed in a customized copper coil that generated AMF (480 A, amplitude 281 kHz frequency) with HOTSHOT 2 (Alonics Co., Ltd., Tokyo, Japan). The change in the temperature of the nanofiber mesh was measured at a predetermined time interval by an FL-IR thermo-camera (CPA-E6, FLIR Systems Japan K.K., Tokyo, Japan). The infrared thermal images of PCL, MNP-PCL, and MNP/DOX/17AAG-PCL nanofiber meshes loaded with 12.0 mg of MNPs under AMF irradiation for 15 min. Additionally, the time-dependent temperature changes of the PCL, MNP-PCL, and MNP/DOX/17AAG-PCL nanofiber meshes and the MNP-PCL nanofiber meshes with different amounts of MNPs during AMF irradiation were analyzed. The sample of nanofiber mesh (40 mg) with 30% (wt) MNPs contained 0.75% (wt) DOX and 17AAG.

### 3.4. In Vitro Drug Release

The release behavior of DOX and 17AAG from the nanofibers was assessed in phosphate buffered saline (PBS) solution at different pH values and with or without AMF irradiation. Nanofiber meshes (0.3 mg DOX, 0.3 mg 17AAG, and 40 mg fiber for each piece) were immersed in 5 mL PBS solution at two different pH values (6.5 and 7.4) by shaking (at 100 rpm) at 37 °C for 30 days. The samples were exposed to AMF for 15 min every three days. At predetermined time intervals, 3 mL PBS was withdrawn and the same volume of fresh PBS was added. Subsequently, the fluorescence (excitation: 485 nm, emission: 595 nm) and absorbance (335 nm) of the collected DOX and 17AAG were measured using a plate reader (Infinite 200PRO, Tecan, Männedorf, Switzerland). The accumulative releases of DOX and 17AAG from the nanofibers were calculated according to their standard curves under PBS conditions.

### 3.5. Evaluation of DOX and 17AAG Efficacy after Heating

MCF-7 cells (human breast cancer cell line) were cultured in MEM supplemented with 10% (*v*/*v*) FBS and 1% penicillin-streptomycin under 5% CO_2_ at 37 °C. MCF-7 cells were plated in a 96-well plate at 1.0 × 10^4^ cells per well and incubated for 24 h. Thereafter, the medium was replaced with fresh culture medium heated to 43 °C and supplemented with DOX or 17AAG at the desired concentration. The cells were incubated for another 24 h after incubation at 43 °C for 0, 5, 15, and 30 min. The culture medium was then removed and incubated in a medium containing 10% (*v*/*v*) Alamar blue assay reagent for 3 h at 37 °C according to a protocol. Finally, the cell number was calculated according to the fluorescence intensity measured by a fluorescent plate reader, and the average value was obtained from six-well measurements. Cell viability was expressed as a percentage of the control culture value. Untreated cells in the growth medium were used as blank controls.

### 3.6. Antitumor Efficacy

MCF-7 cells were cultured in MEM supplemented with 10% FBS and 1% penicillin-streptomycin at 37 °C and 5% CO_2_. MCF-7 cells were plated in a 96-well plate at 1.0 × 10^4^ cells per well for 24 h. Subsequently, the medium was replaced with a preheated medium (43 °C) containing DOX or 17AAG at different concentrations, and the cells were placed into a 43 °C incubator for continuous heating for different time intervals. After the cells were incubated for another 24 h, the fresh medium containing 10% Alamar blue assay reagent was incubated for 3 h at 37 °C. Subsequently, the cell number was calculated according to the fluorescence intensity of the Alamar blue assay measured by a fluorescent plate reader, and the average value was obtained from the six-well measurements.

For the synergistic anticancer experiment, MCF-7 cells were plated in a 35 mm plate at 10^6^ cells per well for 24 h. Thereafter, the medium introduced a piece of nanofiber mesh (which had a 120 mg weight, 30% MNPs, 0.75% DOX, and 0.75% 17AAG), which was exposed to AMF (480 A, amplitude 281 kHz frequency) for 30 min. After incubation at 37 °C for another 24 h, the cell number was calculated by measuring the fluorescence of the Alamar blue assay.

### 3.7. Statistical Analysis

All experiments were performed three times and the data are presented as means ± standard deviation (SD). Statistical analysis was conducted using Student’s *t*-test and one-way analysis of variance (ANOVA) using Origin version 9.0 software (Northampton, USA). The difference between the results was considered to be statistically significant for *p* < 0.05 (*) and *p* < 0.01 (**).

## 4. Conclusions

In this study, we fabricated a novel PCL-based nanofiber mesh with MNPs, DOX, and 17AAG for achieving a mutual synergy of hyperthermia, chemotherapy, and molecularly targeted thermal therapy for a highly potent therapeutic effect. The nanofiber mesh was fabricated using an electrospinning method and exhibited good biocompatibility. The sustained and pH-sensitive release behavior of DOX and 17AAG from the nanofiber for one month was favorable for the long-term maintenance of effective drug concentration in tumor tissue. Meanwhile, the anticancer effect of DOX was enhanced in combination with hyperthermia and 17AAG and displayed a time-dependent relationship. In MCF-7 cells, the MNPs/DOX/17AAG nanofiber mesh efficiently induced apoptosis through the synergistic anticancer effect arising from hyperthermia, DOX, and 17AAG. The results of the study indicate a new tumor therapy methods with high efficiency are critical to maximize the desired therapeutic effect and to minimize the injuries and side effects. Thus, the MNPs/DOX/17AAG nanofiber mesh serves as an effective locally implantable system for enhancing the efficacy of combination cancer treatments.

## Data Availability

Not applicable.

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
