# Peer review of "A Smart Hyperthermia Nanofiber-Platform-Enabled Sustained Release of Doxorubicin and 17AAG for Synergistic Cancer Therapy"

_ijms, 2021, doi:10.3390/ijms22052542_

Round 1

Reviewer 1 Report

This is an interesting contribution reporting the combined effect of hyperthermia and chemotherapy on a MCF-7 cell line showing an enhanced effect the presence of an heat shock protein inhibitor (17AAG). The paper is reasonably well written but there are some errors that I have marked in an annotated pdf file (made with okular software). 

Author Response

Response: We would like to thank the reviewer for appreciating our work. Since we could not find the annotated PDF files, we are now asking the editor. I firmly believe that this will greatly improve our manuscript with valuable comments for revising and improving our paper.

Reviewer 2 Report

It is a well written paper although novelty is rather limited. All the materials and methods have been exploited earlier. I definitely miss the reference to similar paper describing implantable photothermal patch nanofiber system based on polycaprolactone (Mater. Chem. B., 2017, 5, 7504.) References to other two papers (Int. J. Nanotechnol. Nanomed., 2016, 1, 1-15; Biomaterials Sci., 2017, 190) would be also welcome.

Author Response

Response: We would like to thank the reviewer for patiently reading through our manuscript and his/her constructive comments. In the revised manuscript, we have improved the references by added the reviewer-recommended meaningful references as [3], [14], [30].

Reviewer 3 Report

Report of Manuscript ijms-1115560

Title: A Smart Hyperthermia Nanofiber-Platform-Enabled Sustained Release of Doxorubicin and 17AAG for Synergistic Cancer Therapy by Lili Chen et al.

Dear Editor and Authors, I have read the manuscript carefully. I find it very interesting and complete.

In this work, the authors focus on the characterization particular nanofiber with different complementary techniques. The nanofiber is composed of biodegradable poly(caprolactone) with doxorubicin, magnetic nanoparticles, and 17AAG.

The paper is well-written, the English is good and I think that this work could be of interest for the field of basic and applicative research of cancer therapy. The paper is technically sound, but it is limited and specific interest. The work is well structured and the proposed goals were achieved.

The manuscript contains new information to justify publication. The methods described comprehensively. The list of references should be improved and modified. The interpretations and conclusions justified by the results. The manuscript appears complete in all its parts.

However, the manuscript should be improved and it will be worth for publication after minor revisions as recommended below.

Abstract - I suggest that authors avoid acronyms in the abstract.

Keyword – I suggest to add “cancer therapy”. “combination therapy” is not necessary.

Lines 33-36 – About magnetic nanoparticle, I suggest increasing these references. This research field is very vast and of great scientific interest. The authors must increase references. I suggest for example to add “https://doi.org/10.3390/nano10101919”; “https://doi.org/10.3390/app10207322” “https://doi.org/10.3390/nano10112310”.

Sections 3.1  – The authors should indicate the purity of the reagents used.

More details on thermographic acquisitions need to be added. This is unclear and deserves improvement.

Conclusions - The conclusions need to be enriched to emphasize the applicability of the results found: this aspect is fundamental to the publication and impact of this manuscript.

Different minor typo-corrections that should be performed.

This reviewer hopes to receive a new and improved version of the manuscript. The results and details must be particularly emphasized.

Author Response

Comment 1: Abstract - I suggest that authors avoid acronyms in the abstract.

Response 1: We would like to thank the reviewer for this good suggestion. In the revised manuscript, we have replaced acronyms with full names in the abstract and mark them in red color.

Comment 2: Keyword – I suggest to add “cancer therapy”. “combination therapy” is not necessary.

Response 2: We thank the reviewer’s suggestion and agreed that it should be much better to replace “combination therapy” instead of “cancer therapy”.

Comment 3: Lines 33-36 – About magnetic nanoparticle, I suggest increasing these references. This research field is very vast and of great scientific interest. The authors must increase references. I suggest for example to add “https://doi.org/10.3390/nano10101919”; “https://doi.org/10.3390/nano10112310”, “https://doi.org/10.3390/app10207322”.

Response 3: We thank the reviewer’s comments. In the revised manuscript, we have improved the references and added the reviewer-recommended references as [4,5,6].

Comment 4: Sections 3.1 – The authors should indicate the purity of the reagents used.

Response 4: We would like to thank the reviewer for pointing out the missing explanation about the purity of the reagents used. In section “3.1. Materials”, we have indicated the purity of the reagents and mark them in red color.

Comment 5: More details on thermographic acquisitions need to be added. This is unclear and deserves improvement.

Response 5: We thank the reviewer for pointing out defects in our method of thermographic acquisitions. In the revised manuscript, we have added more details to make it more clearly.

Comment 6: Conclusions - The conclusions need to be enriched to emphasize the applicability of the results found: this aspect is fundamental to the publication and impact of this manuscript.

Response 6: In the revised manuscript, we have improved the conclusions by emphasizing the applicability with the data of drug sustained release and the enhancement of chemotherapy effects with hyperthermia and HSP90 inhibitor.